# Trunk Angle Modulates Feedforward and Feedback Control during Single-Limb Squatting

**DOI:** 10.3390/jfmk6040082

**Published:** 2021-10-07

**Authors:** Kristin A. Johnson, Shojiro Nozu, Richard K. Shields

**Affiliations:** 1Department of Physical Therapy and Rehabilitation Science, Roy J. and Lucille A. Carver College of Medicine, The University of Iowa, Iowa City, IA 52242, USA; Kristin-a-johnson@uiowa.edu; 2Institute of Health and Sports Science & Medicine, Juntendo University, Inzai City 270-1695, Japan; s.nozu.hw@juntendo.ac.jp

**Keywords:** motor control, perturbation, reflex, central set, knee, soleus

## Abstract

Trunk positioning and unexpected perturbations are high-risk conditions at the time of anterior cruciate ligament injury. The influence of trunk positioning on motor control responses to perturbation during dynamic performance is not known. We tested the influence of trunk position on feedforward and feedback control during unexpected perturbations while performing a novel single-limb squatting task. We also assessed the degree that feedforward control was predictive of feedback responses. In the flexed trunk condition, there were increased quadriceps (*p* < 0.026) and gluteus medius long-latency reflexes (*p* < 0.001) and greater quadriceps-to-hamstrings co-contraction during feedforward (*p* = 0.017) and feedback (*p* = 0.007) time bins. Soleus long-latency reflexes increased more than 100% from feedforward muscle activity regardless of trunk condition. Feedforward muscle activity differentially predicted long-latency reflex responses depending on the muscle (R^2^: 0.47–0.97). These findings support the concept that trunk positioning influences motor control responses to perturbation and that feedback responses may be invariant to the feedforward control strategy.

## 1. Introduction

The US population continues to be burdened by anterior cruciate ligament (ACL) injuries and the high costs associated with recovery. From 2005–2013 the average cost in healthcare utilization after ACL reconstruction was $13,403.38 with a total of 229,446 surgeries performed [1] and expenses totaling over $3 billion. Numerous biomechanical and motor control risk factors are associated with injury [2,3,4] but screening remains unable to predict injury [5,6], injury rates continue to rise [7], and recurrence after surgical repair is common [8]. The majority of ACL injuries occur by non-contact mechanisms [2] with the knee near full extension [4], often when landing or pivoting [2,4], decelerating [3], or responding to an unexpected event, commonly referred to as a perturbation [2].

A perturbation elicits a rapid update to a pre-planned movement strategy in order to accommodate altered task demands. For example, when an athlete encounters an unexpected perturbation (movement by an opponent; foot contacting the ground prematurely), the pre-planned (feedforward) movement strategy is modified by reflex and volitional (feedback) responses. In most cases these corrective feedback responses facilitate successful completion of the altered motor task. In other cases, such as in non-contact ACL injury, feedback activation of the athlete’s own muscle forces [9] may contribute to injury. Insights into both feedforward and feedback mechanisms can help shed light on the underlying causes of non-contact ACL injury.

Feedback responses to perturbations include the short-latency reflex (0–50 ms), the long-latency reflex (51–200 ms), and volitional reaction time (>200 ms) [10]. The long-latency reflex is the first feedback response with cortical involvement [11] and it occurs at a latency and amplitude that may facilitate a corrective response before soft tissue damage occurs (200 ms) [12]. The long-latency reflex is also task dependent [12] and is modulated by both background muscle activity at the time of the perturbation and the velocity of the imposed stretch [13]. Thus, the long-latency reflex is a window into the nervous system where the simplest neuromuscular behavior overlaps with voluntary control [14]. Importantly, latency [15], amplitude [16], and muscle recruitment sequencing [15] of the long-latency reflex are distinct between individuals with knee dysfunction and healthy controls. We previously developed a method to safely expose people to unexpected knee flexion perturbations while performing a scored motor control task [17,18], demonstrating that quadriceps long-latency reflexes were 40% greater in females with ACL reconstructed knees (~4 years after surgery) compared to healthy controls [16]. Unique to this motor control squatting task is the quantification of human performance. By scoring the task, individuals are encouraged to continue performing in spite of the perturbation as opposed to arresting their movement. This may more closely approximate athletic scenarios when individuals attempt to maximize performance even while responding to unexpected events. Individual differences in long-latency reflexes may contribute to the risk for ACL injury but may also present an intervention opportunity, as they are responsive to alterations in the initial feedforward conditions.

Feedforward factors such as biomechanical position [19,20,21,22], activation of the muscles prior to the perturbation [10,13], and the magnitude of the perturbation [13,18] all determine the initial conditions for a long-latency reflex response. Body position, such as a flexed trunk, is a key way to manipulate the feedforward background muscle activity level [23,24,25]. An upright or extended trunk increases knee joint moments as the center of mass (COM) moves posterior to the base of support [19,21] altering the muscle activation patterns around the knee. Conversely, a flexed trunk shifts the COM anterior, causing a distinctly different pattern of muscle activation that some believe reduces the risk of a non-contact ACL injury [23,24,25,26,27].

A flexed trunk position increases hamstrings activation and posterior shear force at the knee [23] which may mitigate anterior tibial translation and ACL strain during forceful quadriceps contraction [28]. When responding to external perturbations in a static posture, co-contraction of the quadriceps and hamstrings mitigates anterior tibial translation [29]. The effect of trunk positioning on feedforward and feedback control, including quadriceps and hamstrings co-contraction, during unexpected knee perturbations under dynamic performance has not been studied. Accordingly, the purpose of this study was to evaluate the effect of sagittal trunk position on feedforward and feedback control during unexpected perturbations while performing a single-leg squatting task. Additionally, we assessed whether feedforward control could predict feedback control during the unexpected perturbation. We hypothesized that greater trunk flexion would increase co-contraction of the quadriceps and hamstrings during feedforward and feedback responses to an unexpected perturbation during the single-leg squatting task.

## 2. Materials and Methods

### 2.1. Participants

Twenty healthy participants (Table 1; 10 males and 10 females; age 23.9 ± 3.0; weight 71.2 ± 16.7 kg; height 174.2 ± 7.4 cm) were included in the study. Participants with a history of lower extremity or lumbar surgery, injury to either lower extremity or to the lumbar spine in the 6 months prior to the study, or neurological disease were excluded. The study was conducted according to the guidelines of the Declaration of Helsinki, and approved by the Institutional Review Board of the University of Iowa (protocol: 201904706; date of approval: 4 January 2019). Written informed consent was provided by all participants.

### 2.2. Study Design

Participants completed two, 9-condition single-leg squat (SLS) tasks, once with an upright trunk (<15°) and once with a flexed trunk (≥30°) on a custom-built device [17]. The testing order for trunk conditions was counter-balanced across subjects. The trunk segment was defined as the distance from the greater trochanter to the acromion and the trunk angle was calculated in reference to the longitudinal axis per a global coordinate system. An audio cue, provided by a customized LabVIEW program (Labview v12.0f3, National Instruments Corp., Austin, TX, USA), notified participants of target trunk angles during the SLS. The signal was triggered at ≥30° during the flexed condition and ≥15° during the upright condition. Participants in the upright trunk condition maintained <15° trunk flexion as they completed 5 squat cycles. Participants in the flexed trunk condition began in an upright/neutral posture, then moved forward into ≥30° trunk flexion as they descended into knee flexion. The audio cue provided confirmation that they had achieved ≥30° trunk flexion. As they returned to knee extension, they returned to an upright/neutral posture, repeating this pattern for 5 squat cycles. If the audio cue was triggered > 1× during the upright condition or <4× during the flexed condition, then the trial was repeated.

### 2.3. SLS Task

The SLS device consists of a knee pad attached to a rack and pinion system with horizontal resistance administered by an electro-magnetic brake [17,18] (Figure 1). Participants stood atop a stool with their dominant limb affixed to the knee pad and their non-dominant limb suspended by their side. Knee dominance was defined as the preferred kicking leg. Two fingertips from each hand rested on the device for stability. The participant’s knee position was projected on a computer screen (user signal) while they tracked a sinusoidal target (target signal) for 5 squat cycles using a customized program (Labview v12.0f3, National Instruments Corp., Austin, TX, USA). Participants were instructed to match their user signal as closely as possible to the target signal while maintaining trunk condition parameters. Two to three practice trials were allowed before each trunk-position test condition.

The 9 conditions of the SLS test varied by levels of resistance, (5%, 10%, and 15% bodyweight (BW)), and target signal velocities (0.2, 0.4, and 0.6 Hz) which corresponds to approximately 10°/s, 20°/s, and 30°/s knee joint flexion velocities. Once per condition, the resistance of the brake was randomly released to 0% BW during knee flexion (never during the first cycle) causing a perturbation or a freefall into flexion. The perturbation was delivered during the initial 1/3 of knee flexion based on the participant’s horizontal knee position. The duration of brake release was 500, 250, and 200 ms for the 0.2, 0.4, and 0.6 Hz velocities, respectively. SLS signals (target signal, user signal, and perturbation signal) were collected at 2000 Hz using an analog to digital data acquisition (DAQ) board (PCI-6221 National Instruments Corp., Austin, TX, USA). Up to 60 s of rest was allowed between each of the 9 conditions. A 5-min seated rest was mandated between two trunk-position conditions.

Performance was assessed via coherence, absolute error, and user error rate. Coherence is the ability of the user to match the velocity of the sinusoidal task with 0 representing a completely non-related signal and 1 representing a linearly matched signal. Absolute error was calculated as the peak absolute difference between the target and user signals. It was calculated for the entire task (averaged absolute error for each condition) and for the feedforward time bin for perturbed cycles (−50–0 ms prior to perturbation). User error rate, defined as the best-fit line from the target signal minus the user signal, was calculated after the perturbation during the short-latency reflex (0–50 ms post-perturbation) and the long-latency reflex (51–200 ms post-perturbation) time bins.

### 2.4. Electromyography (EMG)

Surface EMG data were collected for the vastus medialis (VM), vastus lateralis (VL), medial hamstrings (MH), lateral hamstrings (LH), gluteus medius (GMed), and soleus (SOL) muscles at 2000 Hz using wireless electrodes (Delsys Trigno, Natick, MA, USA) placed according to SENIAM guidelines [30]. Prior to the SLS, three maximum voluntary isometric contractions (MVIC) were collected in each muscle group with 1-min rest between attempts. The peak MVIC was calculated as the average root mean square EMG for 200 ms on either side of the peak EMG from the three trials. Quadriceps and hamstrings MVIC were collected in a seated position at 90° hip flexion and 60° knee flexion with the distal shank attached to the locked arm of a Biodex System 3 (Biodex Medical Systems, Shirley, NY, USA). SOL MVICs were collected in a seated position with 60° knee flexion and 0° ankle flexion with the foot affixed to a footplate on the Biodex. GMed MVICs were collected in side-lying with a bolster positioned between the legs for 20° hip ABD, 0° hip flexion, and 0° hip medial/lateral rotation. The leg was stabilized to the table with straps proximal to the knee joint and participants attempted to abduct the hip against the resistance of the straps.

EMG and SLS signals were analyzed using a custom DIAdem script (National Instruments, v.2012). Root mean square of the EMG signals were calculated every 10 ms and peak activity was reported as a percentage of the peak MVIC for the feedforward (−50–0 ms), short-latency reflex (0–50 ms), and long-latency reflex (51–200 ms) time bins during the SLS task.

### 2.5. Kinematics

Sagittal kinematics were calculated for the trunk (in reference to the longitudinal axis), hip, knee, and ankle joints during the SLS task using eight motion capture markers and a ten-camera system (Vicon Motion Systems, Inc., Centennial, CO, USA). Kinematics were collected at 100 Hz and synced to the SLS and EMG signals via a DAQ board (PCI-6221 National Instruments Corp., Austin, TX, USA). Frontal and transverse plane motions were not analyzed as the SLS device confines movement to the sagittal plane.

### 2.6. Statistical Analyses

Distinct motor control strategies exist between the sexes [29,31,32]. To account for potential sex differences our data were first analyzed by mixed effects ANOVAs using sex and trunk condition as factors. When there were no significant sex-by-trunk condition interactions or sex main effects, separate one-way repeated measures ANOVAs were used to compare EMG, kinematics, and SLS performance between the flexed and upright trunk conditions. An ANCOVA was used to compare short- and long-latency reflex activity between conditions while adjusting for feedforward muscle activity (−50–0 ms). Simple regression analysis was used to predict short- and long-latency reflex EMG from feedforward EMG for each muscle. Data analysis was conducted on the average of the 10% and 15% BW SLS conditions for each participant. Previous work has shown greater SLS task difficulty at the 10% and 15% BW conditions [18] so the 5% BW conditions were excluded for more robust analysis. We averaged results from six total SLS conditions (10% and 15% BW at 0.2, 0.4, 0.6 Hz velocities each) in each trunk position to best approximate a range of movements people experience in real life. The quadriceps to hamstrings (Q/H) ratio was calculated as the average of the VM and VL peak EMG over the average of the MH and LH peak EMG. Analyses were conducted for perturbed (average of all cycles with a perturbation) squats during the feedforward, short-latency reflex, and long-latency reflex time bins. We had >0.80 power to detect a 10% change in VM, VL, and GMed long-latency reflex activity as a result of trunk angle. Significance level was set at *p* < 0.05. Significant findings were analyzed using Tukey’s post hoc tests.

## 3. Results

There were no differences between male and female kinematics of the trunk, hip, knee, or ankle at the moment when the perturbation was delivered, or during feedforward, short-latency reflex, or long-latency reflex time bins (all *p*’s > 0.05) with the exception of knee flexion velocity during the short-latency reflex (interaction *p* = 0.028, sex *p* = 0.365, trunk condition *p* < 0.001) in which males flexed the knee 44.3°/s and 29.8°/s and females flexed the knee 36.7°/s and 32.6°/s during the flexed and upright trunk conditions, respectively. Analysis of the trunk conditions showed that maximum trunk flexion differed significantly between the two trunk condition tasks (40.3° flexed, 9.6° upright, F(1.39)  =  693.457, *p* < 0.001). At the time the perturbation was delivered, there was 19.3°, 22.8°, and 3.5° greater flexion at the trunk (F(1.39)  =  203.190, *p* < 0.001), hip (F(1.39)  =  114.014, *p* < 0.001), and knee (F(1.39)  =  6.081, *p* = 0.023) in the flexed condition compared to the upright condition. Ankle dorsiflexion at the time of perturbation did not differ between the two trunk conditions (F(1.39)  =  0.811, *p* = 0.379). Figure 2 illustrates a representative example of SLS signals (target and user) and raw EMG during perturbed squat cycles in the flexed and upright conditions.

### 3.1. Effect of Trunk Position on Feedforward Control (−50–0 ms)

There were no differences in feedforward muscle activity between perturbed and unperturbed squat cycles within the same trunk condition for any muscle (all *p*’s > 0.05) suggesting participants did not anticipate the perturbation. Females demonstrated greater feedforward muscle activity than males by 14% (%MVIC) in VM (Figure 3A; interaction *p* = 0.536, sex *p* = 0.020) and by 7.8% (%MVIC) in GMed (Figure 3B; interaction *p* = 0.423, sex *p* = 0.004). There were increases in feedforward muscle activity in the flexed condition for GMed (Figure 3B; *p* < 0.001) and MH (Figure 3C; F(1.39)  =  12.290, *p* = 0.002) compared to the upright condition, but no feedforward muscle activity difference for VM (Figure 3A; *p* = 0.051), VL (Figure 3D; F(1.39)  =  1.306, *p* = 0.267), LH (Figure 3E; F(1.39)  =  1.267, *p* = 0.274), or SOL (Figure 3F; F(1.39)  =  2.085, *p* = 0.165). The quadriceps to hamstrings muscle activity ratio was 45 percent greater in the upright condition versus the flexed condition (Figure 4A; F(1.39)  =  6.846, *p* = 0.017). The ratios of VM to MH (Figure 4B; F(1.39)  =  4.201, *p* = 0.054) and VL to LH (Figure 4C; F(1.39)  =  2.294, *p* = 0.146) muscle activity were not different between trunk conditions.

### 3.2. Effect of Trunk Position on Short-Latency Reflex (0–50 ms)

During the short-latency reflex, females demonstrated 15.1% (%MVIC) and 7.7% (%MVIC) greater muscle activity in the VM (Figure 3A; interaction *p* = 0.776, sex *p* = 0.004) and GMed (Figure 3B; interaction *p* = 0.724, sex *p* = 0.010), respectively. The short-latency reflex was 6–8 percent greater in the flexed versus the upright condition for GMed (Figure 3B; trunk condition *p* = 0.010) and MH (Figure 3C; F(1.39)  =  20.624, *p* < 0.001). There were no differences in the short-latency reflex between trunk conditions for VM (Figure 3A; trunk condition *p* = 0.182), VL (Figure 3D; F(1.39)  =  0.270, *p* = 0.609), LH (Figure 3E; F(1.39)  =  2.732, *p* = 0.115), or SOL (Figure 3F; F(1.39)  =  1.344, *p* = 0.261). The quadriceps to hamstrings muscle activity ratio was 52 percent greater (Figure 4A; F(1.39)  =  12.588, *p* = 0.002), the VM to MH ratio was 85 percent greater (Figure 4B; F(1.39)  =  8.034, *p* = 0.011), and the VL to LH ratio was 33 percent greater (Figure 4C; F(1.39)  =  10.900, *p* = 0.004) in the upright condition versus the flexed condition. When feedforward muscle activity (−50–0 ms) was used as a co-variate, the MH remained 3 percent greater (F(1.37)  =  5.290, *p* = 0.027) and the VL to LH ratio was 53 percent lower (F(1.37)  =  4.508, *p* = 0.04) in the flexed condition versus the upright condition. Feedforward EMG predicted the short-latency reflex response with R^2^ values between 0.75–0.98 for the VM, MH, VL, GMed, LH, SOL, quadriceps to hamstrings, VM to MH, and VL to LH ratios (all *p*’s < 0.001).

### 3.3. Effect of Trunk Position on Long-Latency Reflex (51–200 ms)

Compared to males, females demonstrated 22.7% greater VM activation (Figure 3A; interaction *p* = 0.322, sex *p* = 0.009) and 7.6% greater GMed activation (Figure 3B; interaction *p* = 0.997, sex *p* = 0.024) during the long-latency reflex. Based on the trunk condition, the long-latency reflex was 9–13.5 percent greater in the VM (Figure 3A; *p* = 0.004), GMed (Figure 3B; *p* < 0.001), MH (Figure 3C; F(1.39)  =  17.778, *p* < 0.001), and VL (Figure 3D; F(1.39)  =  5.790, *p* = 0.026), during the flexed condition versus the upright condition. There were no differences in the long-latency reflex between conditions for LH (Figure 3E; F(1.39)  =  2.824, *p* = 0.109) or SOL (Figure 3F; F(1.39)  =  1.112, *p* = 0.305). The time to peak long-latency reflex was shorter during the upright trunk condition compared to the flexed trunk condition in GMed (Figure 3B; interaction *p* = 0.173, sex *p* = 0.617, trunk condition *p* = 0.048). Time to peak long-latency reflexes did not differ between males and females or between trunk conditions for any other muscle (all *p*’s > 0.089) with latencies between 106–154 ms. The quadriceps to hamstrings ratio was 47 percent greater (Figure 4A; F(1.39)  =  9.065, *p* = 0.007), the VM to MH ratio was 60 percent greater (Figure 4B; F(1.39)  =  11.945, *p* = 0.003), and the VL to LH ratio was 36 percent greater (Figure 4C; F(1.39)  =  7.415, *p* = 0.014) in the upright condition versus the flexed condition. Although the SOL long-latency reflex response did not differ between trunk conditions, muscle activity increased more than 100 percent from feedforward background activity (F(1.19)  =  77.734, *p* < 0.001). Using feedforward muscle activity as a co-variate, MH and GMed remained 3.4 (F(1.37)  =  5.251, *p* = 0.028) and 4 percent (F(1.37)  =  5.651, *p* = 0.023) increased in the flexed condition versus upright condition. The VM to MH ratio also remained 27 percent (F(1.37)  =  5.119, *p* = 0.03) greater in the upright condition than the flexed condition. Feedforward EMG predicted the long latency reflex response with R^2^ values between 0.47–0.97 for the VM (Figure 5A), MH, VL (Figure 5B), GMed (Figure 5C), LH, SOL (Figure 5D), quadriceps to hamstrings, VM to MH, and VL to LH ratios (all *p*’s < 0.001).

### 3.4. SLS Performance

Females demonstrated 35.9% greater absolute error during the feedforward time bin than males (interaction *p* = 0.239, sex *p* = 0.044, trunk condition *p* = 0.252) with no other effects of sex on SLS performance (All *p*’s > 0.075). Absolute error was greater throughout the SLS task during the flexed condition versus the upright condition for both the perturbed (F(1.39)  =  12.077, *p* = 0.003) and unperturbed squat cycles (F(1.39)  =  15.832, *p* < 0.001). Coherence was 9 percent greater in the upright condition than in the flexed condition (F(1.39)  =  11.518, *p* = 0.003). During squat cycles with a perturbation, absolute error during the feedforward time bin was no different between trunk conditions (F(1.39)  =  1.364, *p* = 0.257), suggesting participants were performing similarly on the task in both trunk conditions at the time the perturbation was delivered. After the perturbation, the user error rate did not differ between trunk conditions during the short-latency reflex (F(1.39)  =  2.19, *p* = 0.155), but was greater during the long-latency reflex in the flexed condition than the upright condition (F(1.39)  =  11.401, *p* = 0.003). The flexed condition also resulted in greater trunk, hip, knee, and ankle flexion velocity than the upright condition during the feedforward, short-latency reflex, and long-latency reflex time bins (all *p*’s < 0.015).

## 4. Discussion

The purpose of this study was to analyze the influence of sagittal trunk position on feedforward and feedback control during unexpected perturbations while performing a novel single-limb squatting task. Additionally, the study aimed to determine whether feedforward conditions were predictive of feedback responses to perturbation. Several key findings emerged: 1. Squatting with a flexed trunk preferentially augmented feedforward muscle activation in the medial hamstrings and gluteus medius; 2. Squatting with a flexed trunk enhanced long-latency responses to perturbation in the quadriceps, medial hamstrings, and gluteus medius; 3. Soleus activation increased more than 100% during the long-latency reflex from feedforward activity, independent of the trunk condition; 4. Feedforward muscle activity predicted feedback responses to perturbation in both trunk conditions. Taken together, these findings suggest that sagittal trunk position alters both feedforward and feedback mechanisms in ways that may affect the motor response to unexpected perturbations.

### 4.1. Sex Differences

Compared to males, females demonstrated greater difficulty tracking the target signal just prior to the perturbation and greater activation of VM and GMed before and after the perturbation. The effect of trunk positioning on knee flexion velocity immediately after the perturbation (0–50 ms) depended upon sex of the participant. These findings support sex distinct responses to unexpected perturbations which coincides with findings from other perturbation studies [31,33]. It is common clinical practice to alter patient squatting technique in order to mitigate risk for lower extremity injury. Our findings suggest that changing an individual’s sagittal trunk position influences muscle activity and dynamic movement differently in females and males. Although, it is not known whether the increased muscle activity or altered knee flexion velocity demonstrated by females results in clinically meaningful effects. Moreover, the large increase in co-contraction of knee musculature (~50%) when flexing the trunk versus keeping the trunk upright was independent of sex and may be more important for mitigating knee injury [23,29]. Taken together, altering trunk position affects muscle activity and kinematics differently between males and females but, importantly, trunk position may be paramount for the mitigation of knee injury regardless of sex.

### 4.2. Feedforward Activation Predicts Long-Latency Response

Feedforward muscle activity (−50–0 ms) predicted feedback reflex responses (0–50 ms; 51–200 ms) after the perturbation in both trunk conditions. Short-Latency reflexes were more strongly predicted by feedforward activity than long-latency reflexes, consistent with the task dependent nature of long-latency reflexes [12]. Compared to an upright trunk, a flexed trunk elicited greater activation of the VM (~40% MVIC) before the perturbation and greater activation of both VM and VL (~70% MVIC) after the perturbation. However, when adjusted to feedforward muscle activity, quadriceps long-latency reflexes were similar across trunk conditions. In other words, quadriceps reflexes were gained proportionally to the muscle activity prior to the perturbation in both trunk conditions [34]. While increased quadriceps reflexes are expected to stop the freefall induced by the perturbation, the magnitude of the increase (~30% MVIC above the feedforward EMG) and peak of the response (~70% MVIC) is notable considering this was a controlled single-leg squatting task. Situations when the nervous system experiences perturbations while moving at faster velocities, such as athletic competition, may result in reflex magnitudes significant enough to induce injury.

### 4.3. Trunk Flexion Elicits Novel Activation Patterns

A novel finding was the triggering of a long-latency reflex in the gluteus medius when squatting with a flexed trunk. This burst in the gluteus medius, a muscle with a major role in frontal plane stability of the pelvis, persisted even after we adjusted for changes in feedforward control. It is conceivable that the reflex was triggered by a stretch of the posterior gluteus medius muscle fibers or by activation of the vestibular system after the perturbation. Others noted that the vestibular system initiates a hip correction strategy to regain balance [33]; thus, there may have been enhanced motor drive to this muscle to stabilize the COM after the perturbation. There was minimal gluteus medius response when squatting with an upright trunk, which is consistent with our previous findings [17]. Importantly, the gluteus medius assists in controlling the lower extremity [31] and opposes knee valgus moments [35]. Enhancing reflex responses in this muscle by flexing the trunk may offer enhanced protection for the knee during an unexpected perturbation.

A new finding was the substantial increase (>100%) in the soleus long-latency reflex that was independent of trunk position and the associated change in feedforward control. This finding indicates that the soleus, a powerful postural muscle, has the capacity to contribute a significant response to unexpected perturbations. Even though the soleus is a single joint ankle plantar flexor, it has been shown to exert significant posterior tibial shear force at the knee during single-leg landing [36] and unanticipated cutting [35]. Moreover, at the time of ACL injury athletes commonly land in a flatfooted position [20], a stance that may inhibit the soleus from producing posterior shear force at the knee. Future work exploring the influence of soleus excitability on reflex responses to perturbation may be important to fully understanding the relationship between soleus muscle control and knee injury mechanisms.

### 4.4. Functional Relevance of Observed Motor Responses

Short-latency reflexes (0–50 ms) are considered to have little functional relevance [11] when responding to unexpected perturbations [10,13], as is supported by the small increases in muscle activity during the short-latency reflexes in this study (Figure 3). However, the long latency reflexes in this study generated large increases in the quadriceps muscle activity during the perturbation (~70% of MVIC), coupled with much lower change in the antagonist hamstring muscle activity (~25% MVIC). This dynamic muscle imbalance around the knee, if it occurs during real-time high velocity perturbations, may be part of the mechanism that tears the ACL. Feedforward biomechanical positioning can induce a taut ACL [37,38] which when coupled with high magnitude reflex responses to an unexpected perturbation, may further increase the risk of injury and/or may contribute to repetitive strain to the ACL that ultimately leads to failure [39]. Perhaps individuals who are at the highest risk of injury are those who repeatedly strain the ACL by poor feedforward control and excessive, unopposed quadriceps long-latency reflex responses to unexpected perturbations. These findings suggest that feedforward and feedback control are important considerations for understanding the mechanisms that contribute to ACL injury.

Co-contraction of the quadriceps and hamstrings was lower (higher Q/H ratio) in the upright trunk condition for both the feedforward and feedback phases surrounding the perturbation. Primary reliance on the quadriceps to control knee movement, termed quadriceps dominance, is a movement strategy associated with ACL injury [31]. This strategy has been observed in female athletes [32] and individuals with acute ACL deficiency [15] during triggered long-latency reflexes. Additionally, greater relative medial (VM to MH ratio) and lateral (VL to LH ratio) quadriceps activation were found when participants squatted with an upright trunk; ratios that are also related to increased risk factors for knee injury [40,41]. Conversely, there was increased co-contraction (lower Q/H ratio) when squatting with a flexed trunk, a pattern previously associated with decreased ACL strain [23]. While increased co-contraction of the quadriceps and hamstrings may have been induced by proximal trunk movement in the flexed condition versus the upright condition, the force generation by the bi-articular muscles would affect both the hip and knee joints. Other work showed that greater quadriceps and hamstrings co-contraction during unperturbed single leg squatting decreased ACL strain [23]. While ACL strain and knee joint shear forces cannot be determined from this study, together these findings suggest that responding to unexpected perturbations with a flexing trunk may trigger a muscle activation strategy that is more protective of knee injury.

### 4.5. Effect of Trunk Flexion on Task Performance

The single limb squatting task in our study was associated with better performance in the upright trunk condition. Participants were better able to match the velocity of the task (coherence) and minimize the disruption of their knee position after the perturbation (error rate) in the upright trunk condition. The findings of improved single-leg squatting performance coupled with greater injury risk factors (decreased co-contraction) is similar to work by others who showed performance and injury mechanisms were inversely related [42]. Perhaps one of the reasons injury screening tests remain insufficient [5,6] and ACL injury rates continue to rise across sport [7] is that individuals adopt “safe” patterns during scripted testing but revert to “unsafe” movement during on-field performance. The novelty of the movement task and device used in this study may assist us in bringing the laboratory closer to the on-field performance. Our capacity to test feedforward and feedback control during a functionally relevant movement while the athlete strives to maximize performance is unique to the SLS task used in this study. While more research is warranted, the SLS task used in this study may ultimately have utility for pre-season screening and/or rehabilitation of the injured athlete.

### 4.6. Study Considerations

It is important to consider that knee injury is often the result of multi-planar forces [3,22], thus, caution should be used when drawing inference about injury from the sagittal plane SLS task. Additionally, the flexed trunk condition may have created a dual-task challenge (tracking the target signal while listening for an audio cue), resulting in greater difficulty of the task [43]. Future work may consider the relationship between cognition, feedforward, and feedback control surrounding unexpected perturbations or attempt to minimize the dual-task aspect of the test.

## 5. Conclusions

In summary, feedforward muscle activation predicted feedback reflex responses after unexpected perturbations during a dynamic single-leg squatting task, independent of biomechanical trunk position. Flexing the trunk while squatting resulted in greater co-contraction around the knee, increased multi-segmental lower extremity joint flexion, and increased feedforward activation and long-latency reflexes in the quadriceps, gluteus medius, and medial hamstrings during unexpected perturbations. Task performance such as velocity matching and knee control were improved when squatting with an upright trunk. Together, these findings suggest performing single-leg squats with concomitant trunk flexion may decrease risk factors associated with ACL injury but at the cost of performance. Clinical interventions to improve feedforward control during movement tasks may be important in modulating responses to unexpected perturbations.

## Figures and Tables

**Figure 1 jfmk-06-00082-f001:**
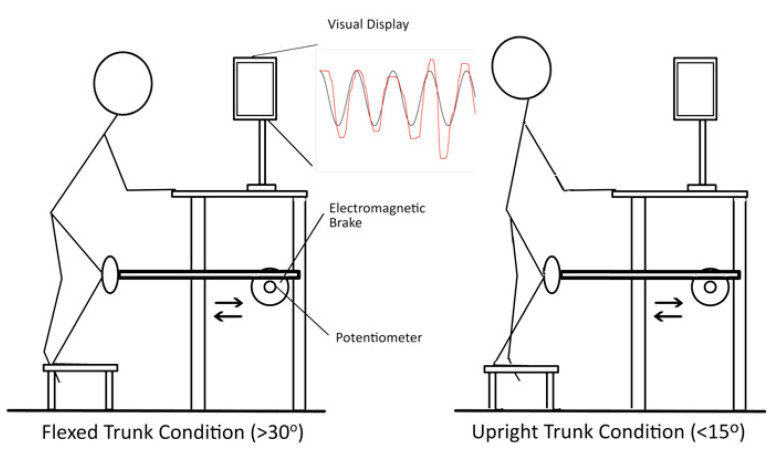
Single-leg squat (SLS) device and experimental conditions. Five single-leg squat cycles were performed under 9 different conditions which varied by resistance at the knee (5, 10, 15% body weight) and velocity (0.2, 0.4, 0.6 Hz) of the target signal. The target signal (black sinusoid wave) was displayed on the computer monitor along with participant knee position, user signal (red sinusoid wave). A random perturbation was elicited once per condition during the descending squat phase. Flexed Trunk Condition: Attain ≥30° trunk flexion once per squat cycle. Upright Trunk Condition: Maintain <15° trunk flexion throughout the squatting task.

**Figure 2 jfmk-06-00082-f002:**
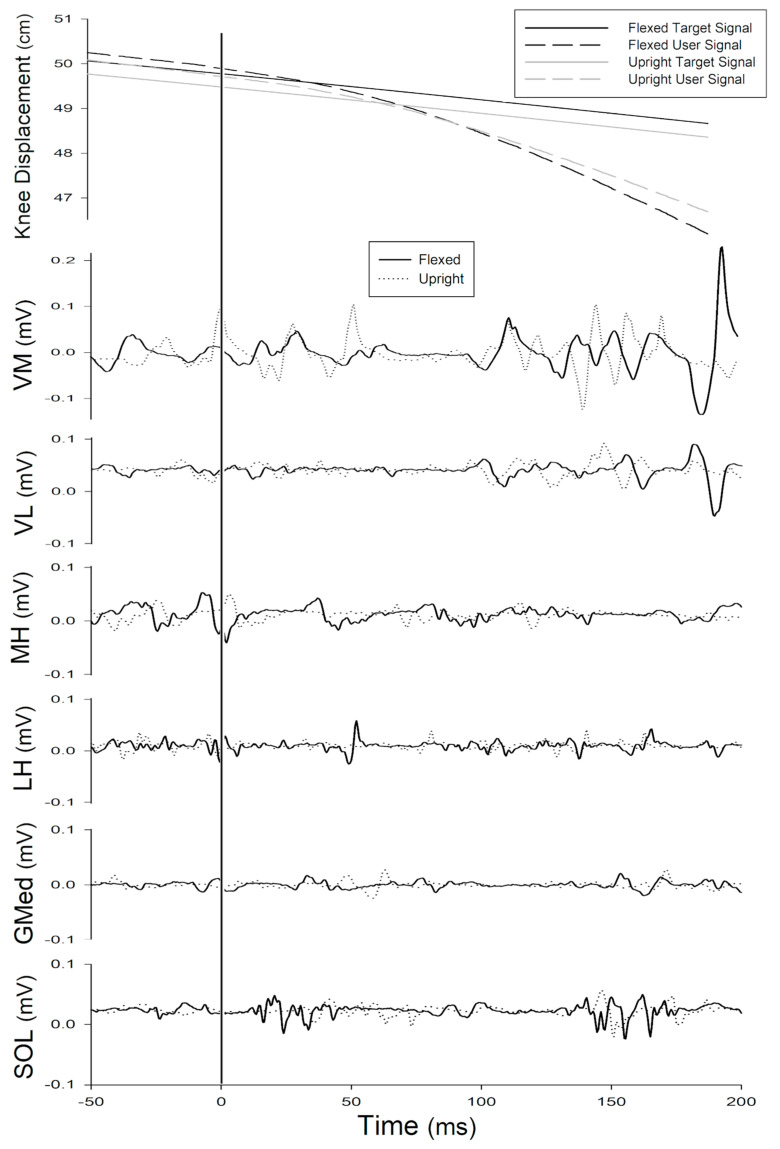
Representative example from a single participant of linear knee displacement (top graph) and raw electromyography (EMG) signals for 250 ms around the perturbation under the flexed and upright trunk conditions. Top graph: solid lines represent the target signal (target knee position) and dashed lines represent the user signal (participant knee position) for the flexed (black) and upright (gray) trunk conditions. Bottom graph: EMG signals from vastus medialis (VM), vastus lateralis (VL), medial hamstrings (MH), lateral hamstrings (LH), gluteus medius (GMed), and soleus (SOL). Solid lines represent EMG in the flexed trunk condition and dotted lines represent EMG in the upright trunk condition. *X*-axis represents time (ms); the brake release occurred at 0 ms.

**Figure 3 jfmk-06-00082-f003:**
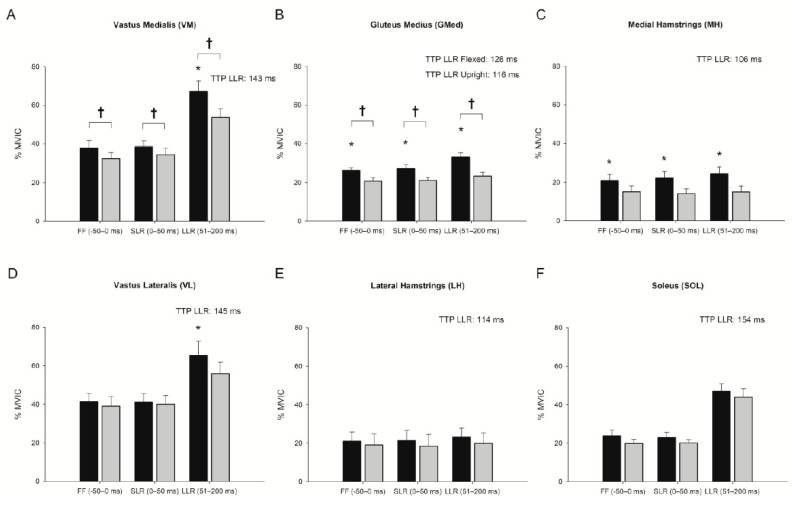
Peak EMG from a perturbed squat cycle during the feedforward (−50–0 ms), short-latency reflex (SLR; 0–50 ms) and long-latency reflex (LLR; 51–200 ms) for: (**A**) vastus medialis (VM); (**B**) medial hamstrings (MH); (**C**) gluteus medius (GMed); (**D**) vastus lateralis (VL); (**E**) lateral hamstrings (LH); and (**F**) soleus (SOL) in the flexed (black bars) and upright (gray bars) trunk conditions. ^†^ Represents significant difference in muscle activity between males and females (*p* < 0.05) * Represents significant difference between trunk conditions (*p* < 0.05). Time to peak (TTP) LLR is displayed for each muscle and represents the average of both trunk conditions except for GMed which differed between trunk conditions (*p* = 0.048).

**Figure 4 jfmk-06-00082-f004:**
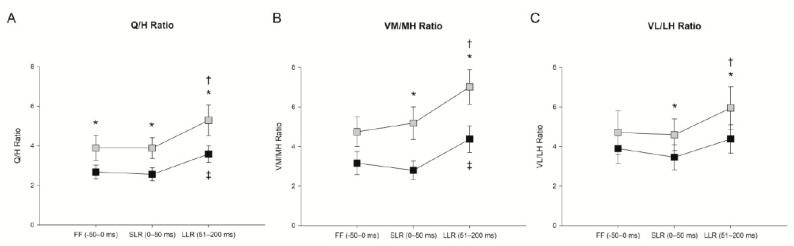
(**A**) Peak quadriceps to hamstrings ratio (Q/H), (**B**) peak vastus medialis to medial hamstrings ratio (VM/MH) and (**C**) peak vastus lateralis to lateral hamstrings ratio (VL/LH) during the feedforward (FF; −50–0 ms), short-latency reflex (SLR; 0–50 ms) and long-latency reflex (LLR; 51–200 ms) for perturbed squat cycles. Black squares represent ratios in the flexed trunk condition. Gray squares represent ratios in the upright trunk condition. * Represents significant difference between trunk conditions (*p* < 0.05). ^†^ Represents significant difference from FF time bin within upright trunk condition (*p* < 0.05). ^‡^ Represents significant difference from FF time bin within flexed trunk condition (*p* < 0.05).

**Figure 5 jfmk-06-00082-f005:**
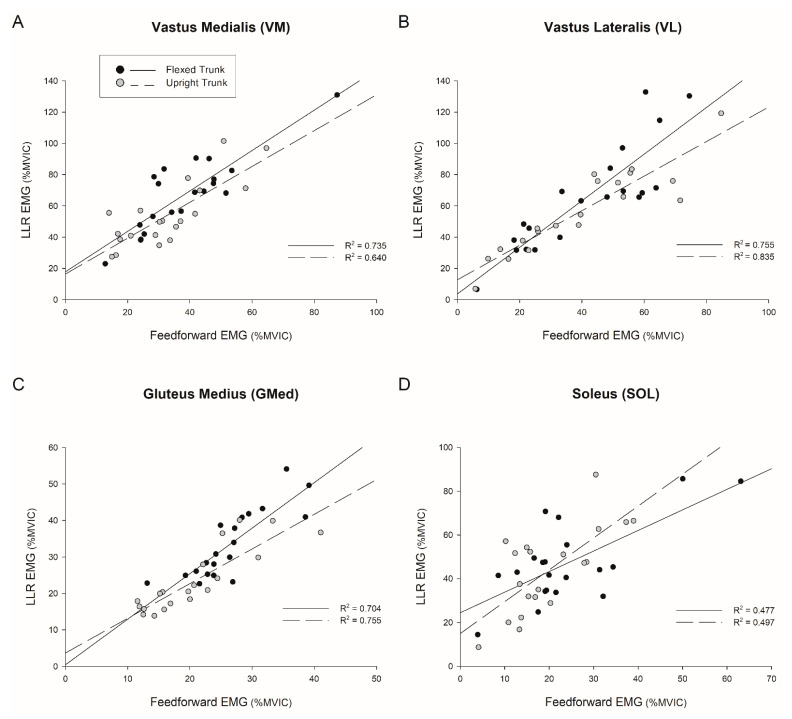
Simple Linear Regression showing the relationship between feedforward and feedback muscle activity during perturbed squat cycles. *X*-axis represents peak feedforward EMG (−50–0 ms) as %MVIC during perturbed squat cycles. *Y*-axis represents peak long-latency reflex EMG (LLR; 51–200 ms) as %MVIC during perturbed squat cycles. Black circles and solid regression lines represent the flexed trunk condition. Gray circles and dashed regression lines represent the upright trunk condition. Regression analysis for (**A**) vastus medialis (VM), (**B**) vastus lateralis (VL), (**C**) gluteus medius (GMed), and (**D**) soleus (SOL). All regressions were significant at *p* < 0.001.

**Table 1 jfmk-06-00082-t001:** Participant characteristics. Values are expressed as mean (standard deviation). Sex differences determined by independent *t*-tests. Significance set at *p* < 0.05.

Variable	Males (*n* = 10)	Females (*n* = 10)	*p*-Value
Age (years)	24.7 (4.1)	23.0 (0.7)	0.210
Height (cm)	176.5 (7.1)	172.0 (7.4)	0.174
Weight (kg)	80.9 (16.1)	61.5 (10.9)	0.006 *

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
