# Peer review of "Trunk Angle Modulates Feedforward and Feedback Control during Single-Limb Squatting"

_jfmk, 2021, doi:10.3390/jfmk6040082_

Round 1

Reviewer 1 Report

This study suggested interesting results and experimental suggestions. 
Also, this paper is well written with logical flow. 
However, I found some major limitations and suggest revision related these issues. 
This paper deserves to be published after major revisions.

page 2. 
A total of 20 subjects participated in the study, 10 males and 10 females each. 
Organize age, height, weight, etc. into a table and add a table so you can check it at a glance. 
Are there any differences between men and women? 
If you think that differences between men and women can affect the results, it is necessary to re-analyze the results.

Author Response

Reviewer 1:

Thank you for your positive comments and helpful insight.  We agree that analyzing the data for sex differences is important.  We performed this analysis and have updated the manuscript accordingly.  We believe these changes make the manuscript stronger and we thank you for your recommendations.

  1. Introduction

[lines 56-61]  Additional context for the reader about the uniqueness of our assessment device.

  1. Methods

2.1

[lines 92-95] We updated our IRB statement to include the required information.

[lines 97-102] We added a Table to visualize participant information per your recommendation.

2.6

[lines 193-196] We analyzed the data using sex as a between subject factor in a Mixed Model ANOVA and have updated the Results due to the new analyses.

  1. Results

[lines 214-220]  Details the sex * trunk condition interaction for knee flexion kinematics during the short-latency reflex.

3.1

[lines 239-242]  New data showing sex effect on FF muscle activity.

[lines 243-246]  Reorganization of data to coincide with order of reporting significant findings.

[lines 252-257]  Figure 3 has been updated to show sex effects and to display GMed before MH to coincide with the order of reporting in the manuscript.

3.2

[lines 259-261] New data showing sex effect on SLR.

[lines 262-264] Reorganization of reported findings

[line 265] updated statistical results based on Mixed Model

3.3

[lines 284-286] New data showing sex effect on LLR.

[lines 287]  Updated statistical findings for VM based on Mixed Model.

[lines 288-290]  Reorganization of reported findings.

[lines 293-299]  Updated statistical findings for time to peak LLR based on sex effect.

3.4

[lines 320-322]  New data showing sex effect for SLS performance during the feedforward time bin.

  1. Discussion

[line 342-345]  Updated discussion based on sex differences in which quadriceps muscle activation was no longer enhanced during the feedforward time bin while flexing the trunk but remained elevated during the long-latency reflex.

4.1 

[lines 352-367]  Added discussion and citations to explain new findings based on sex effects.

4.6

[lines 463-467]  Eliminated the lack of sex analysis as a limitation as this data is now included in the Results.

Reviewer 2 Report

Dear Authors,

Congratulations for the work done. The text is well written and methodologically the research is correct.

However, the authors should contextualize better in the Introduction and discuss more deeply the results obtained in relation to the importance of trunk movement in postural control.

It is  necessary that the authors improve the contextualization of the assessment instruments used and discuss more deeply the implication of the results obtained.

Kind regards.

Author Response

Reviewer 2:

Thank you for your kind feedback and your recommendations.  In the Introduction we cover a range of topics to provide context for and the necessity of each of the outcomes under investigation in this study.  We agree that the squatting task is not explained in detail in this section, however, we briefly describe the motor control task in lines [53-56] and provide citations to our other studies for further detail on this task (citations 16-18).  We have added more information in lines [56-61] on the uniqueness of this type of motor control task to improve the context for its use.  We believe that the additional comments [lines 56-61], the details in the Methods section, and our reference to other studies now provide sufficient information for readers to fully understand the task that was performed.

We have expounded upon potential implications of our results in detail throughout the Discussion.  We explain the influence of trunk positioning on motor control in section 4.3 and 4.5 and provide additional relevance of the findings in section 4.4.

Round 2

Reviewer 1 Report

Before the revision of the paper, a total of 22 subjects were randomly assigned to 2 groups of 11 subjects. After revision, the subjects were divided into 20 male and female groups. Why were two people excluded?
Because there is a difference between men and women, it seems normal to have differences in height and weight. It is doubtful whether the two subjects in question were excluded to make the age similar.

Author Response

We are confused by this statement.  In all versions of this manuscript we have reported 20 participants.  It was never 22.  It was always 20 with 10 females and 10 males.  We therefore did not exclude any participants or their data.  While it may be common to find height differences between the sexes, the males and females in our cohort were similar in height and age but different in weight as reported in all submissions.